# Effect of Freeze–Thawing Treatment on Platelet-Rich Plasma Purified with Different Kits

**DOI:** 10.3390/ijms25189981

**Published:** 2024-09-16

**Authors:** Ryoka Uchiyama, Haruka Omura, Miki Maehara, Eriko Toyoda, Miyu Tamaki, Makoto Ogawa, Tatsumi Tanaka, Masahiko Watanabe, Masato Sato

**Affiliations:** 1Department of Orthopaedic Surgery, Surgical Science, Tokai University School of Medicine, 143 Shimokasuya, Isehara 259-1193, Japan; uchiyama.ryoka.m@tokai.ac.jp (R.U.); omura.haruka.w@tokai.ac.jp (H.O.); maehara.miki.e@tokai.ac.jp (M.M.); toyoda.eriko.r@tokai.ac.jp (E.T.); miyu.tamaki90@gmail.com (M.T.); ogawa.marina.g@tokai.ac.jp (M.O.); tanaka.tatsumi.g@tokai.ac.jp (T.T.); masahiko@tokai.ac.jp (M.W.); 2Center for Musculoskeletal innovative Research and Advancement (C-MiRA), Tokai University Graduate School, 143 Shimokasuya, Isehara 259-1193, Japan

**Keywords:** platelet-rich plasma, autologous protein solution, osteoarthritis of the knee, macrophage polarization

## Abstract

Osteoarthritis of the knee (OAK), a progressive degenerative disease affecting quality of life, is characterized by cartilage degeneration, synovial inflammation, and osteophyte formation causing pain and disability. Platelet-rich plasma (PRP) is an autologous blood product effective in reducing OAK-associated pain. PRP compositions depend on their purification. In clinical practice, PRP is typically administered immediately after purification, while cryopreserved PRP is used in research. Platelets are activated by freezing followed by release of their humoral factors. Therefore, PRP without any manipulation after purification (utPRP) and freeze–thawed PRP (fPRP) may differ in their properties. We purified leukocyte-poor PRP (LPPRP) and autologous protein solution (APS) to compare the properties of utPRPs and fPRPs and their effects on OAK target cells. We found significant differences in platelet activation and humoral factor content between utPRPs and fPRPs in both LPPRP and APS. Freeze–thawing affected the anti-inflammatory properties of LPPRP and APS in chondrocytes and synovial cells differed. Both utPRPs and fPRPs inhibited polarization toward M1 macrophages while promoting polarization toward M2 macrophages. Freeze–thawing specifically affected humoral factor production in macrophages, suggesting that evaluating the efficacy of PRPs requires considering PRP purification methods, properties, and conditions. Understanding these variations may enhance therapeutic application of PRPs in OAK.

## 1. Introduction

Osteoarthritis of the knee (OAK) is a chronic degenerative disease characterized by abnormal cartilage metabolism, osteophyte formation, and synovial inflammation. OAK leads to pain, disability, and a significant decline in quality of life [1]. The number of patients with OAK is increasing worldwide, but no treatment currently exists that leads to a complete recovery from OAK [2,3]. The intraarticular injection of platelet-rich plasma (PRP) has attracted attention and is rapidly spreading as a safe and simple treatment because it has low allergic risk and no need for complicated processes, such as cell cultivation [4].

As an autologous blood product, PRP contains various growth factors and cytokines [5]. The popularity of PRP treatment has led to the proposal for a range of purification methods and it has been reported that the composition of blood components and humoral factors in PRP varies depending on the purification method and kit used [6,7,8].

Abundant in PRP, platelets are known for their storage of growth factors and cytokines that play a crucial role in regulating tissue healing [9]. Platelets are easily activated in response to physical and chemical stimuli, and they release their content humoral factors, which exert an influence on the proinflammatory/anti-inflammatory and anabolic/catabolic processes into the surrounding environment [9]. The abundant presence of anti-inflammatory factors and growth factors is considered to contribute to symptom improvement in OAK by inhibiting inflammation and promoting tissue repair [10,11,12]. Clinically, PRP is injected immediately after purification during treatment, but previous basic studies have usually used cryopreserved PRP [13,14,15]. Recent studies have shown that centrifugation speed and storage condition affect platelet activity and the humoral factor level [16,17,18]. The purification process and storage of PRP may affect platelet activity, which affects humoral factor content and may result in different therapeutic effects, but this has not been investigated.

In this study, we intended to investigate the anti-inflammatory effects and macrophage polarization induced by different platelet-rich plasma (PRP) preparations. We used two PRP purification kits and purified leukocyte-poor PRP (LPPRP) and autologous protein solution (APS) from the peripheral blood of healthy subjects. Then, part of each PRP was frozen at −80 °C. We compared the platelet activation rate and humoral factor concentration in respective PRPs, which we then added into the culture media of chondrocytes and synovial cells to investigate their anti-inflammatory effects. Additionally, we added the respective PRPs into the culture media of monocyte-derived macrophages (MDMs) and M1 macrophages to investigate their effect on macrophage polarization.

## 2. Results

### 2.1. Changes in Platelet Activation and Humoral Factor Concentration through Freeze–Thawing of PRPs

First, we examined changes in platelet activation rate and humoral factor concentration during the preparation and freeze–thawing of PRP. Eighty milliliters of peripheral blood was collected from each healthy donor and LPPRP and APS were purified. Hematologic analysis was performed on whole blood, LPPRP, and APS immediately after purification without freezing (Table 1). There was no significant difference in platelet concentration between LPPRP and APS.

The positive rate for the CD40L and CD62P platelets was measured in whole blood (WB), plasma after first centrifugation (1st), PRP without any manipulation after purification (utPRPs), and PRP frozen at −80 °C and thawed at different temperatures (4 °C, 20 °C, and 37 °C) by flow cytometry (Figure 1).

In the CD40L-positive rate, significant differences were observed only in LPPRP, where utPRPs exhibited a higher activation rate compared with WB and the first container. No effect of PRP freeze–thawing was observed in either LPPRP or APS. (Figure 1b,d).

In the CD62P-positive rate, no significant difference was observed in the preparation process for either LPPRP or APS. The comparison of utPRPs and platelets activated by freezing (fPRPs) showed that activation of fPRPs was greater than that of utPRPs in both LPPRP and APS, regardless of the thawing temperature (Figure 1c,e). The activation rate tended to increase with thawing temperature; PRPs thawed at 37 °C were used as fPRPs for subsequent experiments.

Humoral factor concentrations in utPRPs (LPPRP, APS) and fPRPs (i.e., frozen LPPRP [fLPPRP] and frozen APS [fAPS], thawed at 37 °C) were quantified using bead-based immunoassays (Table 2).

Comparing PRPs and fPRPs, nine factors in LPPRPs and five factors in APS significantly changed in concentration through freeze–thawing. Interleukin-13 (IL13), interleukin1 receptor antagonist (IL1RA), basic fibroblast growth factor (bFGF), epidermal growth factor (EGF), platelet-derived growth factor-AA (PDGF-AA), PDGF-BB, and vascular endothelial growth factor (VEGF) in LPPRP and IL-13, tumor necrosis factor receptor-1 (TNF-R1), monokine induced by interferon-γ (MIG), and granulocyte-macrophage colony-stimulating factor (GMCSF) in APS were significantly higher in fPRPs compared with utPRP. However, IL-1RA and fLPPRP exhibited significantly higher concentrations than LPPRP; for TNF-R1, LPPRP and fAPS exhibited higher concentrations than fLPPRP and APS; and for TGFβ, the concentration was higher in utPRP compared to fPRPs, respectively.

The results suggest that platelets in PRP are activated by freeze–thawing and humoral factor concentration changes.

### 2.2. Effect of Freeze–Thawing of PRPs on Osteoarthritis

To evaluate the effect of freeze–thawing on the anti-inflammatory effect of PRP in OAK, synovial cells and chondrocytes obtained from total knee arthroplasty waste tissue were stimulated with IL1β. Subsequently, respective PRPs were added to the synovial cells and chondrocytes. The effects on gene expression were assessed using qualitative reverse transcription polymerase chain reaction (qRT-PCR), and humoral factor production was measured using enzyme-linked immunosorbent assay (ELISA). Protein production of IL12 and TNFα were low in detection.

In synovial cells, both LPPRP and fLPPRP significantly decreased IL6 and IL12 gene expression and matrix metalloproteinase-13 (MMP13) protein production compared to the control. However, fLPPRP significantly reduced the protein production of IL6 and MMP13 compared to LPPRP (Figure 2a,b). APS and fAPS also significantly decreased IL6 and IL12 gene expression compared with the control, with APS showing a more significant reduction than fAPS. Conversely, fAPS significantly increased MMP13 gene expression over APS, and APS and fAPS increased IL6 protein production over the control, respectively (Figure 2c,d).

In chondrocytes, both LPPRP and fLPPRP significantly decreased the gene expression of IL6, MMP3, MMP13, and the protein production of MMP13 compared to the control. Gene expression of IL6 was significantly reduced in fLPPRP compared to LPPRP, but no other significant differences were observed between LPPRP and fLPPRP (Figure 3a,d). APS and fAPS decreased IL6 and MMP13 gene expression and MMP13 protein production compared to the control. APS showed significantly lower MMP13 gene expression than fAPS, and a similar trend was observed for IL6 gene expression. Furthermore, APS decreased MMP-3 gene expression, while fAPS increased MMP-3 protein production (Figure 3c,f).

Concerning the expression of cartilage-related genes, both LPPRP and fLPPRP significantly decreased the expression of Col1 and ACAN, and APS and fAPS significantly reduced the expression of Col1, ACAN, and SOX9 compared to the control, with no difference observed between PRP and fPRP (Figure 3b,e).

These results suggest that freeze–thawing of PRP affects the expression of inflammatory factors and MMPs in synovial cells and chondrocytes. Specifically, in LPPRP, fPRP tended to downregulate the expression of these factors more effectively than utPRP. Whereas in APS, utPRP downregulated the expression of inflammation-related factors more effectively than fPRP. The effect of freeze–thawing of PRP on its anti-inflammatory properties appears to differ between LPPRP and APS.

### 2.3. Effect of Freeze–Thawing on the Indirect Effects of PRP on Osteoarthritis

Recently, macrophages in periarticular connective tissues have been reported to be involved in the progression of OAK. In the synovial membrane and fluid of patients with OAK, alternatively activated macrophages (i.e., M2 macrophages), which have the ability to repair tissues, are decreased, while classically activated macrophages (i.e., M1 macrophages), which promote inflammation, are increased [19,20,21].

To investigate the effect of freeze–thawing of PRP on macrophage polarization, we added each PRP to MDM and M1 macrophages, which were differentiated from monocytes isolated from the peripheral blood of healthy donors, for 48 h. The M1/M2 macrophage phenotype markers were evaluated using flow cytometry and ELISA assays.

To investigate the effect for macrophage polarization, phenotypic analysis of MDMs after the addition of LPPRP and fLPPRP showed that both LPPRP and fLPPRP significantly decreased the expression of CD80 and the production of TNFα, an M1-associated marker, compared to the control. However, no significant difference was observed between LPPRP and fLPPRP. The expression of CD86 was significantly upregulated in fLPPRP compared to both the control and LPPRP. Whereas protein production of IL10, an M2-associated marker, tended to be upregulated in LPPRP, no significant difference was observed (Figure 4b,c). The expression of phenotypic markers in MDMs after the addition of APS and fAPS tended to decrease CD80 expression and TNFα production, while tending to increase CD163 and CD206 gene expression, compared to the control. However, no significant difference was observed between APS and fAPS. Additionally, CD86 expression was significantly decreased in the fAPS compared to the control and APS, while IL10 protein production was significantly increased in the APS compared to fAPS (Figure 4d,e).

To investigate the effect on M1 macrophage polarization, analysis of the M1 phenotype using cell surface markers after the addition of LPPRP and fLPPRP showed that fLPPRP significantly upregulated CD86 expression compared to both the control and LPPRP. In contrast, the expression of M2-associated markers tended to increase in both LPPRP and fLPPRP. Additionally, IL10 protein production tended to be upregulated in both LPPRP and fLPPRP compared to control, with LPPRP showing a stronger tendency, but no significant difference was observed (Figure 5a,b). Regarding APS and fAPS, fAPS significantly upregulated CD86 expression compared to the control, while tending to decrease TNFα protein production. In contrast, CD163 expression tended to increase in both APS and fAPS, but no significant difference was observed. CD206 was downregulated by APS compared to the control and fAPS, whereas IL10 protein production increased compared to the control. However, none of these differences were significant (Figure 5c,d).

These results suggest that both LPPRP and APS inhibit polarization to M1 macrophages and promote polarization to M2 macrophages in MDMs and M1 macrophages. Additionally, freeze–thawing of PRP may primarily affect the production of humoral factors.

## 3. Discussion

In this study, we demonstrated that freezing changes the platelet activation rate and humoral factor content of PRP, resulting in differential effects on target cells. Moreover, the impact of freeze–thawing on target cells depends on the PRP properties.

OAK develops and progresses by the upregulation of inflammation and cartilage degradation driven by catabolic factors such as inflammatory cytokines and matrix metalloproteinases [22]. PRP therapy for knee joints may reduce pain and other symptoms through balancing inflammation and cartilage destruction by the action of anti-inflammatory factors and growth factors abundant in PRP [11,12]. While many studies support the efficacy of PRP, there are also reports of negative outcomes [23,24,25]. It is important to note that the PRP used in these studies often differed in purification methods, activation treatments, and cryopreservation protocols [12,26,27].

Platelets are known to easily activate and release humoral factors in response to various stimuli; the properties of PRP used in basic research may differ from those used in clinical practice. This discrepancy suggests that the results observed in basic studies may not directly translate to clinical outcomes. However, the impact of these differences has not yet been verified.

In this study, we used two types of PRP (i.e., LPPRP and APS) that were each purified using different clinical purification kits to investigate the effects of freeze–thawing on platelet activation and humoral factor content. We also evaluated the effect of freeze–thawing on the anti-inflammatory effects of PRP on synovial cells and chondrocytes, and its effect on macrophage polarization.

Consistent with previous studies, CD62P values, which indicate platelet activation rates, were significantly higher in fPRP, with a tendency for an increased activation ratio associated with higher thawing temperatures (Figure 1) [17]. Although it has been reported that platelets are activated by centrifugation speed, no significant differences in platelet activation rates were observed for both LPPRP and APS during the preparation process [28]. One reason for this finding is that the protocol recommends dissolving the activated and aggregated platelets by standing or gently tapping to subside platelets to maximize platelet collection. Although, the blood cell composition of LPPRP and APS is different, both PRP showed a similar tendency of platelet activation through freeze–thawing, suggesting that the blood cell components do not significantly affect platelet activation through this process (Figure 1). Quantification of humoral factors revealed significant differences between PRP and fPRP observed both LPPRP and APS. Consistent with previous studies, TGF-β were decreased in fPRP compared to utPRP [29,30]. In this study, we only measured free active TGF-β, and freezing may have reduced its activity, contributing to the observed decrease. Even in LPPRP, which contains leukocytes, IL1RA, IL4, IL13, and the other leukocyte-derived factors showed higher concentrations in fPRP than in utPRP. It is possible that freeze–thawing also stimulated the erythrocytes and leukocytes in the PRP [31]. These results suggest that freeze–thawing may stimulate blood cells contained in PRP, changing the contents of the humoral factors that can affect efficacy (Table 2).

We evaluated the anti-inflammatory effects of PRP using synovial cells and chondrocytes isolated from discarded tissue during total knee arthroplasty. We found that LPPRP showed significantly lower MMP13 production in synovial cells, and significantly higher IL6 production in both synovial cells and chondrocytes compared to fLPPRP (Figure 2a,b and Figure 3a,c). Conversely, APS showed significantly higher levels of IL6 and MMP13 production in synovial cells, and significantly lower levels of MMP13 expression in chondrocytes compared to fAPS (Figure 2c,d and Figure 3d,f). These results showed the importance of considering PRP purification methods and properties, including freezing and activation, when evaluating PRP efficacy. For chondrocyte-related gene expression, similar results were observed for Col1, ACAN, SOX9, and RUNX2 in both utPRP and fPRP, indicating that freezing does not affect the anabolic/catabolic functions of PRP on chondrocytes (Figure 3b,e). Although there is a discrepancy between gene expression and humoral factor production, protein production is regulated by various post-transcriptional processes, and mRNA and protein expression levels do not always correlate [32]. While previous studies have reported decreased humoral factors with longer culture periods, the 48-h culture period used in this study was too short to adequately evaluate [14,33]. Furthermore, while all measured proteins are known to be present in PRP, the possibility of PRP remaining in the wells cannot be excluded.

Some reports suggest that the therapeutic effects of PRP, which may persist beyond the duration of platelet presence in the joint following administration, are indirectly influenced by macrophages in the synovial tissue [34]. Macrophages are immune cells involved in tissue homeostasis and immune responses, and they can polarize into different phenotypes based on their microenvironment [35]. Classically activated macrophages (i.e., M1 macrophages) associated with tissue damage and inflammation. Conversely, alternatively activated macrophages (i.e., M2 macrophages) are associated with tissue repair and anti-inflammatory responses [36].

Comparing the concentrations of humoral factors involved in macrophage polarization between utPRP and fPRP, we found that M1- polarization factors, such as IFNγ and TNFα, were higher in utPRP, while M2- polarization factors like IL4, IL10, and IL13 were higher in fPRP (Table 2). In both MDM and M1 macrophages, the addition of PRP tended to maintain or decrease M1 macrophage markers while increasing M2 macrophage markers; these findings are consistent with previous studies [37]. Our results suggest that both utPRP and fPRP inhibit polarization toward M1 macrophages while promoting polarization toward M2 macrophages. Additionally, freeze–thawing of PRP appears to specifically affect the production of humoral factors in macrophages. Although M1-associated factors may increase, this effect may be counterbalanced by a corresponding increase in M2-associated factors.

Limitations of our study include small sample sizes and short storage time. We purified PRP using peripheral blood from healthy subjects, Wasai et al. reported that the humoral factor composition of PRP purified from healthy subjects differed from that of PRP from OAK patients [27]. In addition, chondrocytes and synovial cells were isolated from patients with grade 4 Kellgren–Lawrence OAK, which means that this study tested the responses to different donor PRPs. There is the possibility that platelets became activated during culture after the addition of the medium, but we did not measure that in this study. Furthermore, approximately 2.5 mL of APS was obtained from 60 mL of peripheral blood, and approximately 2.0 mL of LPPRP from 20 mL of peripheral blood per kit. Thus, while the amount of PRP injected into the joint during clinical practice differs between LPPRP and APS, the total amount of humoral factors injected into the joint may also differ, potentially leading to different clinical outcomes. However, the same amount of PRP was added to the culture medium in this study. Finally, due to the absence of in vivo studies and histological analyses, in vitro studies have limitations in their ability to predict crosstalk between tissues.

Future research should investigate the differences in component profiles associated with various PRP preparation methods and their therapeutic efficacy, particularly with respect to freeze–thawing. Also, in clinical research, select OAK patients, based on the findings of this study, inject PRP, so the intra-articular condition and the degree of improvement in symptoms should be evaluated. Such studies are crucial for establishing optimal PRP preparation methods and their clinical applications, leading to enhancement of the efficacy and appropriate use of PRP in the treatment of OAK.

In conclusion, our study demonstrated that freeze–thawing on PRP stimulated blood cells in the PRP, leading to changes in humoral factor concentrations. These changes likely contribute to differences in anti-inflammatory effects on chondrocytes and synovial cells, as well as alterations in macrophage polarization. These findings could enhance the efficacy, dissemination, and appropriate use of PRP therapy.

## 4. Materials and Methods

### 4.1. Ethics Statement

This study was reviewed and approved by the Institutional Review Board of the Tokai University School of Medicine (21R-306, 23R-064, 23R-085) and was conducted in compliance with relevant guidelines. Written informed consent was obtained from all participants.

### 4.2. PRP Purification

To purify PRPs, 80 mL of peripheral blood was collected from 6 healthy subjects (M = 2, F = 4, age = 38.6 ± 11.0 years), and was added to 8 mL of anticoagulant citrate-dextrose solution A (ACD-A; TERUMO, Tokyo, Japan).

LPPRP was purified using a CellAid^®^ Serum Collection Set P type kit (JMS, Hiroshima, Japan). This kit consists of primary and secondary containers connected at the top by multiple tubes. Blood (20 mL) containing ACD-A was injected into the primary container and centrifuged at 200× *g* for 15 min. The plasma layer containing platelets was transferred to the secondary container via a tube at the top and centrifuged at 1200× *g* for 15 min. Excess plasma was returned to the primary container, the pelletized platelets were disrupted by tapping, and then approximately 2 mL of LPPRP was collected.

APS was purified using an APS kit (Zimmer Biomet, Warsaw, IN, USA). This kit consists of two independent tubes (a GPS III system and an APS Separator). Blood (60 mL) containing ACD-A was injected into the cell separation tube (GPS III system) and centrifuged at 745× *g* for 15 min using a dedicated centrifuge (Zimmer Biomet) and 6 mL of the upper layer (PRP layer) was collected. This 6 mL was added to the APS Separator and centrifuged at 219× *g* for 2 min in the same centrifuge and approximately 2.5 mL of APS was collected.

Half of the volume of the collected PRPs was immediately used for the experiment, while the other half of the PRPs were immediately frozen at −80 °C for 30 min (fPRPs). The fPRPs were thawed at 4 °C, 20 °C (room temperature), and 37 °C until completely thawed for the analysis of platelet activation. For the addition to cell culture, the frozen PRPs were thawed at 37 °C.

### 4.3. Hematologic Analysis of PRPs

The leukocyte, erythrocyte, and platelet concentrations of WB, LPPRP, and APS were determined using an automated hematology analyzer (XT-1800i; Sysmex, Kobe, Japan) immediately after preparation.

### 4.4. Cell Isolation and Culture of Chondrocyte and Synovial Cells

A modified version of a previously reported protocol was used [38,39]. Chondrocyte and synovial cells were isolated from cartilage and synovial tissue obtained from five patients (M = 1, F = 4, Age = 71.4 ± 7.8 years) who underwent total knee arthroplasty at Tokai University Hospital.

Briefly, cartilage and synovial tissue were separately minced and subsequently digested with 5 mg/mL collagenase type I (Worthington Biochemical Corp., Lakewood, NJ, USA) in minimum essential medium α (αMEM; Gibco, Waltham, MA, USA) supplemented with 10% fetal bovine serum (FBS; SAFC Bioscience, Lenexa, KS, USA) and 1% antibiotic, antimycotic solution (AB; Fuji Film, Tokyo, Japan). After 4 or 2 h, respectively, cells were filtered through a 100-μm cell strainer (BD Bioscience, San Diego, CA, USA) and washed in Dulbecco’s phosphate-buffered saline (DPBS). Primary chondrocytes were stored at −80 °C in Cell Banker 1 cryopreservation medium (Zenoaq, Fukushima, Japan), and synovial cells were seeded at 1 × 10^4^ cells/cm^2^ and cultured to Passage 1 and then cryopreserved similarly.

Stored synovial cells (Passage 1) and chondrocytes (Passage 0) were thawed and seeded in flasks at 1 × 10^4^ cells/cm^2^ or 5 × 10^4^ cells/cm^2^, respectively. The cells were cultured until subconfluence, then seeded in 24-well plates and 96 well plates at 1 × 10^4^ cells/cm^2^ and cultured again until subconfluence. The cells were then used for the experiment (synovial cells Passage 2 and chondrocytes Passage 1). The medium was then replaced with αMEM supplemented with 10% FBS, 1% AB, and 10 ng/mL IL1β (Peprotech, Rocky Hill, NJ, USA), and cultured for 24 h. After that, the medium was replaced with αMEM supplemented with 10% FBS and 1% AB, and 10% volume of each PRP was added to the 0.4 μm pore inserts, and then cultured for another 2 days.

For gene expression analysis, cells were lysed, and cDNA was synthesized from RNA using a SuperPrep II cell lysis and RT kit for qPCR (TOYOBO, Osaka, Japan) according to the manufacturer’s protocol. To measure the concentration of humoral factors, the medium was changed to αMEM supplemented with 1% AB, and the culture supernatants were collected after 24 h. The supernatants were then centrifuged at 15,885× *g* for 5 min to remove debris and stored at −80 °C until analysis. The control group (*n* = 5) consisted of 5 synovial cells or chondrocyte donors, and the PRP group (*n* = 30) consisted of 5 synovial cells or chondrocyte donors, with 6 PRP donors per group.

### 4.5. Isolation and Culture of Monocyte-Derived Macrophage and M1 Macrophages

A modified version of a previously reported protocol was used [36,40,41]. Peripheral blood mononuclear cells (PBMCs) were isolated from the buffy-coat of five healthy donors (M = 2, F = 3, Age = 35.8 ± 13.7 years) using a density gradient (Histopaque 1077, Sigma-Aldrich, St. Louis, MO, USA). The PBMCs were washed with wash buffer containing DPBS and 1% FBS, and then centrifuged at 540× *g* at 4 °C for 5 min. To remove contaminating red blood cells, a red blood cell lysing buffer (Sigma-Aldrich) was added and incubated for 10 min at 37 °C, followed by another centrifugation step at 540× *g* at 4 °C for 5 min. The resulting cell pellet was resuspended, and Fc receptor (FcR) blocking reagent (Miltenyi Biotech, Bergisch Gladbach, Germany) was added to prevent nonspecific antibody binding through FcRs. The cells were then incubated with anti-human CD14 microbeads (Miltenyi Biotech) at 4 °C for 20 min. After washing with wash buffer, CD14+ monocytes were isolated using an autoMACS^®^ Pro Separator (Miltenyi Biotech).

Collected monocytes were seeded at a density of 1 × 10^5^ cells/cm^2^ on an Upcell Multi 24-well plate (CellSeed, Tokyo, Japan) and 96-well plates (Thermo Fisher Scientific, Tokyo, Japan) with Roswell Park Memorial Institute 1640 media containing GlutaMAX^TM^ supplement (RPMI1640; Gibco), supplemented with 10% heat-inactivated FBS, 1% AB, and 20 ng/mL MCSF (Peprotech), and incubated at 37 °C and 5% CO_2_.

For evaluating the effects on macrophage polarization, after 6 days of culture with M-CSF, the medium was replaced with RPMI1640 medium supplemented with 10% heat-inactivated FBS and 1% AB, and 10% volume of each PRP was added on 0.4 μm pore inserts. The cells were then cultured for another 2 days.

For evaluating the effects on M1 macrophage polarization, after 6 days of culture with MCSF, the medium was replaced with RPMI 1640 medium supplemented with 10% heat-inactivated FBS, 1% AB, 50 ng/mL IFNγ, and 100 ng/mL LPS and the cells were cultured for 2 days to polarize into M1 macrophages. Then, the medium was replaced with RPMI1640 medium supplemented with 10% heat-inactivated FBS and 1% AB, and 10% volume of each PRP was added on 0.4 μm pore inserts and cultured for another 2 days.

To measure the concentration of humoral factors, the medium was changed to αMEM supplemented 1% AB, and the culture supernatants were collected after 24 h, centrifuged 15,885× *g* for 5 min to remove debris, and stored at −80 °C until analysis. The control group (*n* = 5) consisted of 5 monocyte donors, and the PRP group (*n* = 30) consisted of 5 monocyte donors, 6 PRP donors per group.

### 4.6. Analysis of Humoral Factors

The concentrations of 24 types of humoral factors (EGF, Fas, Fas ligand [FasL], FGF basic, granzyme B, GMCSF, HGF, IFNγ, IL1RA, IL4, IL6, IL8, IL10, IL13, inducible protein 10 (IP10), MCSF, MIG, PDGF-AA, PDGF-BB, free activeTGFβ, TNFα, TNF-R1, TNF-R2, and VEGF) in LPPRP, fLPPRP, APS, and fAPS were measured simultaneously using a flow cytometry bead-based immunoassay (LEGENDplex™, BioLegend, San Diego, CA, USA) according to the manufacturer’s protocol. All PRPs were filtered through a 1-μm cell strainer (pluriSelect Life Science, Leipzig, Germany) without any external activation. Data were acquired using a FACS Verse™ Flow Cytometer (BD Bioscience) and analyzed using cloud-based LEGENDplex™ Data Analysis Software (BioLegend).

To measure the humoral factors in the culture supernatants, ELISA kits were used to measure the concentration of MMP3 and MMP13 (Abcam, Cambridge, UK) and a cytometric bead array was used for IL6, IL10, IL12, and TNFα (BD Bioscience). The assays and analysis were performed following the manufacturer’s protocols. Data were normalized using the CellTiter-Glo^®^ 2.0 Cell Viability Assay (Promega, Madison, WI, USA) and measured with Glomax (Promega).

### 4.7. Flow Cytometry

To analyze the platelet activation rates in peripheral blood and PRPs, the samples were divided into two tubes. In one tube, the samples were reacted with the following four mouse monoclonal anti-human antibodies for multiple staining (i.e., CD40 ligand APC [Clone: 89-76], CD41a PE-Cy7 [Clone: HIP8], CD61 FITC [Clone: VI-PL2], and CD62P PE [Clone: AK-4]; BD Bioscience). Samples in the other tube were mixed with nonspecific fluorescent mouse IgGs as a negative control. Both tubes were reacted at room temperature for 20 min in the dark. Then, 1% paraformaldehyde (4% paraformaldehyde phosphate buffer solution (Fujifilm) diluted with DPBS was added and the samples were incubated at 4 °C for 1.5 h in the dark. Then, the samples were washed, and data were acquired.

To analyze the macrophage phenotypes, the Upcell Multi 24-well plates were kept at room temperature for 30 min to promote detachment of the cells, and the cells were collected with cold FACS buffer (DPBS + 1% BSA) by pipetting the contents of each well. FcR blocking reagent (Miltenyi Biotech) was added to the cell suspension and the cell suspension was kept at room temperature for 15 min to inhibit Fc receptor-mediated nonspecific antibody binding.

The cell suspension was then divided into two tubes. In one tube, the cells were reacted with the following four mouse monoclonal anti-human antibodies for multiple staining (i.e., CD80 PE-Cy7 [Clone: L307.4], CD86 APC [Clone: 2331], CD163 FITC [Clone: GHI/61], and CD206 PE [Clone: 19.2]; BD Bioscience). Cells in the other tube were mixed with nonspecific fluorescent mouse IgGs as a negative control. Both tubes were reacted at 4 °C for 30 min in the dark. The cells were washed and then incubated with Cytofix Fixation buffer (BD Bioscience) for 20 min at 4 °C. The cells were then washed again, and data were acquired.

Data acquisition was performed using a BD FACS Verse™ Flow Cytometer (BD Bioscience) and the data were analyzed using FlowJo (Tree Star, Ashland, OR, USA).

### 4.8. Gene Expression Analysis

cDNA was used for qRT-PCR. The qRT-PCR analysis was performed using TaqMan^®^ Fast Advanced Master Mix (Applied Biosystems) on a QuantStudio^®^ 3 Real-Time PCR System (Applied Biosystems). The PCR cycling conditions were as follows: 50 °C for 2 min, 95 °C for 2 min, followed by 40 cycles of 95 °C for 1 s and 60 °C for 20 s. The probes used for qRT-PCR included GAPDH (Hs02758991_g1), IL6 (Hs00985639_m1), IL12 (Hs00168405_m1), TNFα (Hs00174128_m1), MMP3 (Hs00968305_m1), MMP13 (Hs00233992_m1), COL1A1 (Hs00164004_m1), COL2A1 (Hs00264051_m1), ACAN (Hs00153936_m1), SOX9 (Hs01001343_g1), and RUNX2 (Hs01047973_m1). Quantitative measurements of all primers used in this study were determined using the 2^−ΔΔCt^ method, and GAPDH expression was used as the internal control.

### 4.9. Statistical Analysis

Numerical results were statistically analyzed using SPSS^®^ statistics software (v. 26; IBM SPSS, Armonk, NY, USA). In this study, due to the use of different cell types, we based our effect size on previous research, setting it at 0.06. The significance level (α) was set at 0.05, and the desired power level was set at 80% (β = 0.2). Dunnett’s test was used to compare against the control group, and a one-way ANOVA test was employed for comparisons among three or more groups. The significance level was set at *p* < 0.05.

## Figures and Tables

**Figure 1 ijms-25-09981-f001:**
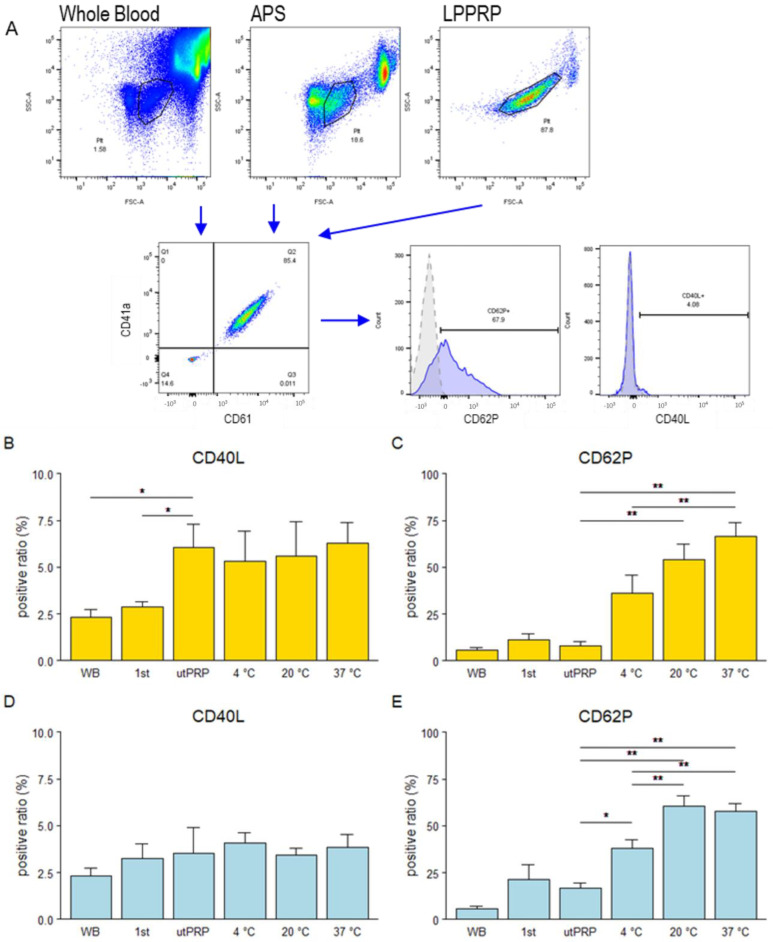
(**A**) Flow cytometric analysis of platelet activation markers. (**B**,**C**) The positivity rates of CD40L (**B**) and CD62P (**C**) in platelets contained in LPPRP. (**D**,**E**) The positivity rates of CD40L (**D**) and CD62P (**E**) in platelets contained in APS. WB = whole blood, 1st = after first centrifugation, utPRP = PRP without any manipulation. Data are presented as the mean ± SEM. * *p* < 0.05, ** *p* < 0.01.

**Figure 2 ijms-25-09981-f002:**
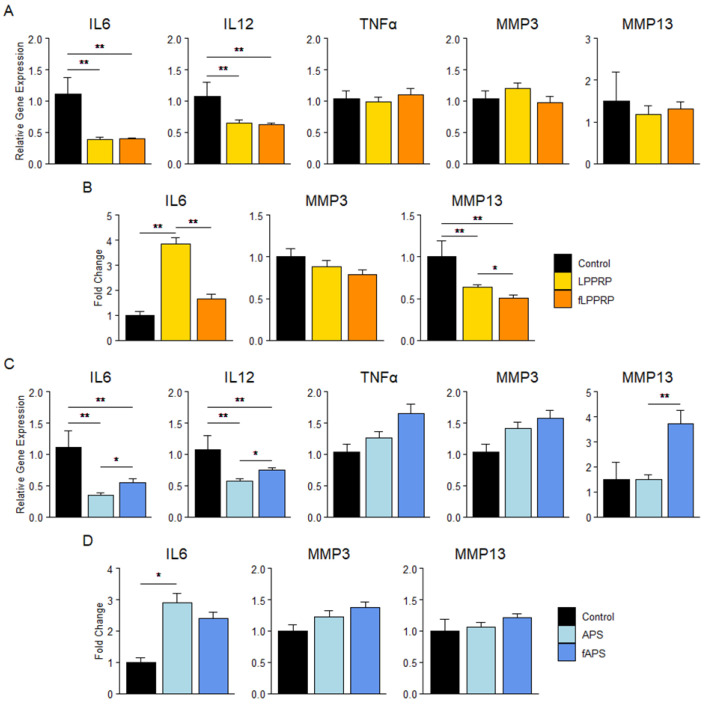
Effect of PRPs on gene expression and humoral factor production of inflammatory cytokines and matrix metalloproteinases in synovial cells. (**A**,**C**) Gene expression of inflammatory cytokines (i.e., IL6, IL12, TNFα) and MMPs (i.e., MMP3, MMP13) in synovial cells treated with LPPRP (**A**) and APS (**C**). (**B**,**D**) Humoral factor concentration in culture media of synovial cells treated with LPPRP (**B**) and APS (**D**), normalized by CellTiter-Glo^®^ assay. Mean ± SEM are indicated. Control group = IL1β-stimulated cells. * *p* < 0.05, ** *p* < 0.01 by one-way analyses of variance (ANOVA). Control group (*n* = 5): 5 synovial cells donors, PRP group (*n* = 30): 5 synovial cells donors, 6 PRP donors per group.

**Figure 3 ijms-25-09981-f003:**
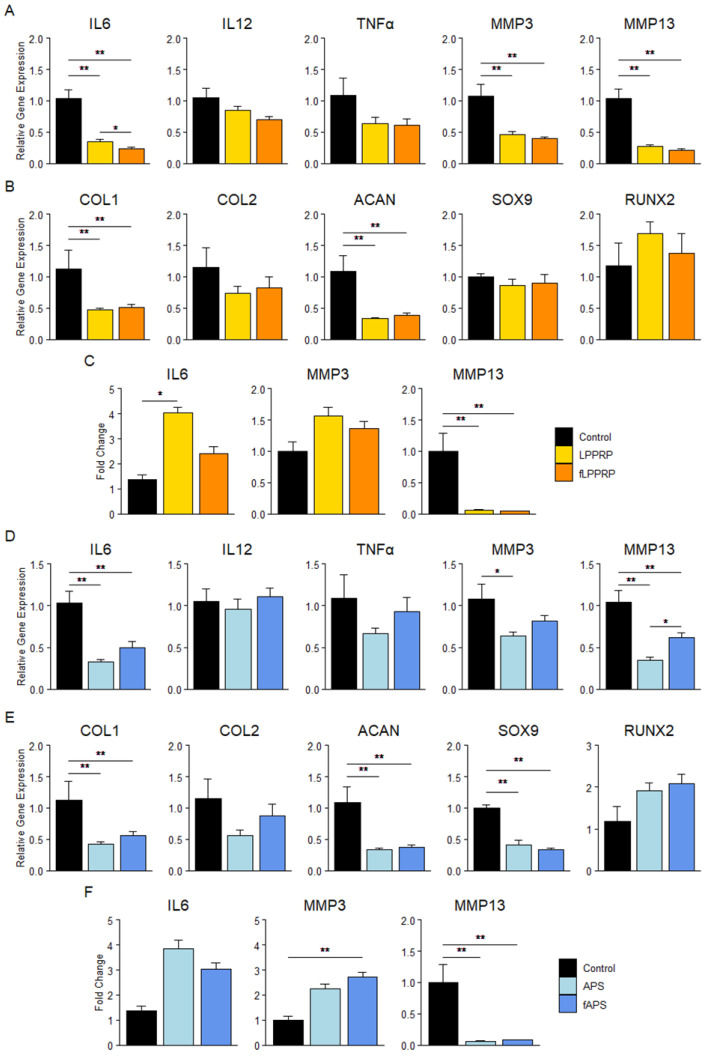
Effect of PRPs on gene expression and humoral factor production of inflammatory cytokines and matrix metalloproteinases in chondrocytes. (**A**,**B**,**D**,**E**) Gene expression of inflammatory cytokines (i.e., IL6, IL12, TNFα), MMPs (i.e., MMP3, MMP13), and cartilage-related genes in chondrocytes treated with LPPRP (**A**,**B**) and APS (**D**,**E**). (**C**,**F**) Humoral factor concentration in culture media of chondrocytes treated with LPPRP (**C**) and APS (**F**), normalized by CellTiter-Glo^®^ assay. Mean ± SEM are indicated. Control group = IL-1β-stimulated cells. * *p* < 0.05, ** *p* < 0.01 by one-way ANOVA. Control group (*n* = 5): 5 chondrocyte donors, PRP group (*n* = 30): 5 chondrocyte donors, 6 PRP donors per group.

**Figure 4 ijms-25-09981-f004:**
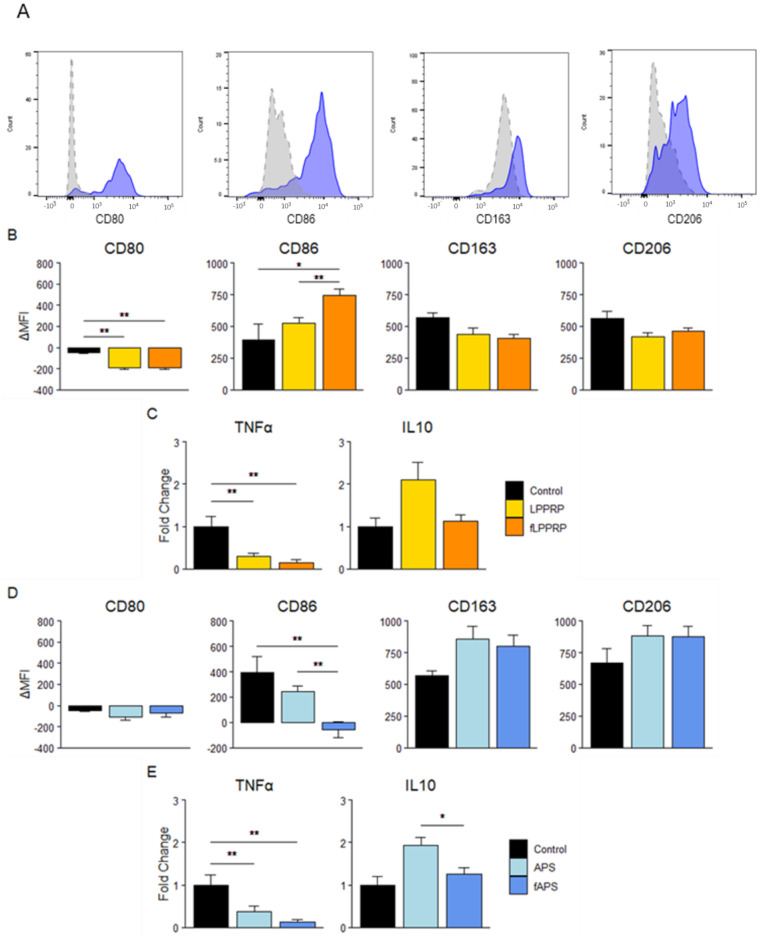
Effect of PRPs on macrophage polarization. (**A**) A histogram representation of typical flow cytometry results analyzing macrophage phenotype. Gray and dashed lines: isotype control; blue and solid lines: signals for each antibody. Mean fluorescence intensity (MFI) values of each antibody were used to calculate ΔMFI values: ΔMFI = MFI Sample–MFI Isotype. (**B**,**D**) Surface levels of CD80, CD86 (i.e., M1-associated markers), CD163, and CD206 (i.e., M2-associated markers) on MDM after treatment with LPPRP (**B**) and APS (**D**). (**C**,**E**) Humoral factor concentration in culture media after treatment with LPPRP (**C**) and APS (**E**), normalized by CellTiter-Glo^®^ assay. Mean ± SEM are indicated. Control group = MDM. * *p* < 0.05, ** *p* < 0.01 by one-way ANOVA. Control group (*n* = 5): 5 monocyte donors, PRP group (*n* = 30): 5 monocyte donors, 6 PRP donors per group.

**Figure 5 ijms-25-09981-f005:**
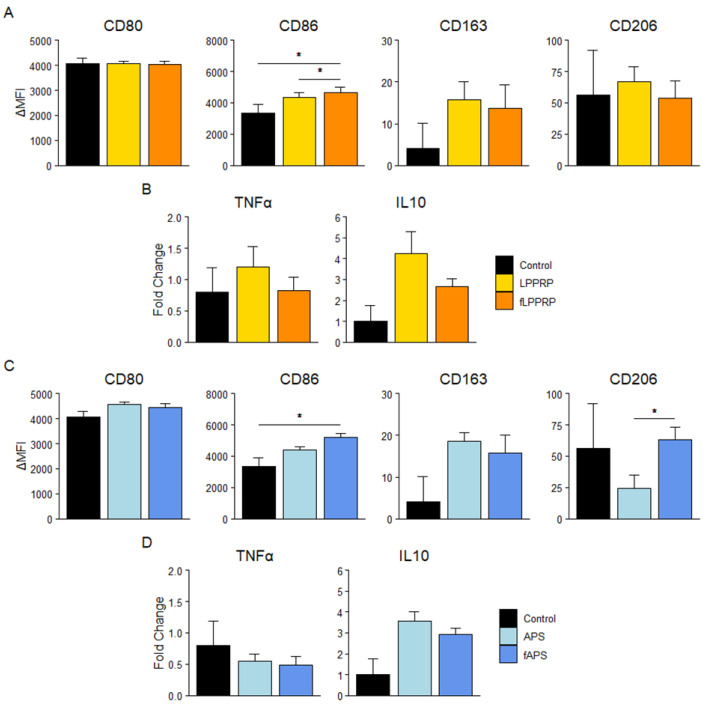
Effect of PRPs on M1 macrophage polarization. (**A**,**C**) Surface levels of CD80, CD86 (i.e., M1-associated markers), CD163, and CD206 (i.e., M2-associated markers) on M1 macrophages after treatment with LPPRP (**A**) and APS (**C**). (**B**,**D**) Humoral factor concentration in culture media after treatment with LPPRP (**B**) and APS (**D**), normalized by CellTiter-Glo^®^ assay. Mean ± SEM are indicated. Control group = M1. * *p* < 0.05 by one-way ANOVA. Control group (*n* = 5): 5 monocyte donors, PRP group (*n* = 30): 5 monocyte donors, 6 PRP donors per group.

**Table 1 ijms-25-09981-t001:** Characteristics of PRP purified using two kits.

	Whole Blood	LPPRP	APS
Leukocytes (×10^2^/μL)	50.5 ± 15.67	2.2 ± 1.12 ^b^	429.6 ± 158.06 ^a,b^
Erythrocytes (×10^4^/μL)	431.9 ± 38.67	2.3 ± 3.61 ^a,b^	262.5 ± 158.00 ^a,b^
Platelets (×10^2^/μL)	19.3 ± 3.24	74.8 ± 37.69 ^a^	54.8 ± 27.25

Hematologic analysis of WB, LPPRP, and APS (*n* = 6). Data are presented as the mean ± standard deviation. ^a^ Significant difference at *p* < 0.05 compared with WB, ^b^ Significant difference at *p* < 0.05 compared with LPPRP and APS.

**Table 2 ijms-25-09981-t002:** Humoral factor concentrations in PRPs.

	LPPRP	fLPPRP	*p*-Value		APS	fAPS	*p*-Value
IL6	4.3 ± 1.0	3.6 ± 0.7	0.317	IL6	5.2 ± 1.3	3.8 ± 0.7	0.090
IL8	59.0 ± 22.5	46.5 ± 13.8	0.370	IL8	77.5 ± 32.1	69.2 ± 20.7	0.684
IFNγ	1011.8 ± 278.5	2924.8 ± 845.0	0.108	IFNγ	1063.5 ± 600.5	835.6 ± 118.7	0.850
TNFα	5.5 ± 3.3	1852.8 ± 572.1	0.185	TNFα	38.5 ± 32.1	651.2 ± 588.0	0.379
IL4	311.7 ± 192.0	1604.7 ± 1141.6	0.341	IL4	267.8 ± 72.9	10,824.0 ± 6942.2	0.490
IL10	526.6 ± 300.3	842.2 ± 345.2	0.555	IL10	11,037.5 ± 6589.2	11,395.5 ± 6687.5	0.302
IL13	1421.9 ± 988.9	24,266.2 ± 4800.3	0.018	IL13	109.0 ± 46.1	24,021.8 ± 5978.2	0.016
IL1RA	1848.6 ± 585.7	16,183.6 ± 4945.6	0.037	IL1RA	15,871.4 ± 8013.7	25,208.3 ± 6046.5	0.463
TNF-R1	1905.6 ± 98.6	1132.7 ± 64.8	0.000	TNF-R1	2473.5 ± 325.7	4410.2 ± 462.9	0.013
TNF-R2	97.4 ± 23.6	70.7 ± 10.6	0.115	TNF-R2	173.4 ± 30.8	168.5 ± 14.9	0.880
bFGF	143.0 ± 24.2	456.8 ± 45.1	0.001	bFGF	182.4 ± 41.6	213.9 ± 80.1	0.686
EGF	22.3 ± 5.4	55.4 ± 7.7	0.004	EGF	19.0 ± 2.5	90.7 ± 19.3	0.010
HGF	79.6 ± 31.2	65.4 ± 23.3	0.197	HGF	189.7 ± 33.6	186.3 ± 25.7	0.903
PDGF-BB	398.2 ± 96.3	1451.8 ± 412.1	0.005	PDGF-BB	528.2 ± 175.3	1530.3 ± 518.6	0.270
PDGF-AA	657.0 ± 123.9	1468.5 ± 281.9	0.021	PDGF-AA	1000.0 ± 162.1	1413.1 ± 308.9	0.088
TGFβ	1089.5 ± 216.2	343.7 ± 54.4	0.012	TGFβ	387.0 ± 283.8	199.3 ± 55.0	0.565
VEGF	11.6 ± 2.8	15.8 ± 2.2	0.040	VEGF	14.9 ± 2.9	23.8 ± 3.6	0.058
GZMB	598.5 ± 142.6	495.8 ± 112.2	0.102	GZMB	721.9 ± 125.6	952.1 ± 155.6	0.201
IP10	1652.6 ± 825.0	2701.0 ± 1179.6	0.175	IP10	933.1 ± 451.1	1223.3 ± 532.1	0.671
MIG	455.2 ± 398.0	5159.0 ± 4801.9	0.474	MIG	641.6 ± 518.3	22,979.1 ± 7020.9	0.045
GMCSF	1774.9 ± 1340.2	5681.3 ± 4205.4	0.731	GMCSF	299.0 ± 75.2	22,982.8 ± 7017.2	0.049
MCSF	69.0 ± 51.0	682.7 ± 430.4	0.374	MCSF	131.6 ± 83.7	15,701.4 ± 8274.3	0.148
Fas	121.3 ± 5.3	122.4 ± 7.4	0.852	Fas	189.0 ± 28.7	203.2 ± 15.0	0.747
FasL	19.8 ± 3.3	19.1 ± 1.8	0.830	FasL	27.4 ± 4.0	23.2 ± 3.2	0.284

Quantification of each humoral factor in LPPRP, fLPPRP, APS, and fAPS purified from healthy donors (*n* = 6). Measured by bead-based immunoassay. Data are presented as the mean (pg/mL) ± SEM.

## Data Availability

The data presented in this study are available on request from the corresponding author. The data are not publicly available because of confidentiality issues.

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
