# Peer review of "Effect of Freeze–Thawing Treatment on Platelet-Rich Plasma Purified with Different Kits"

_ijms, 2024, doi:10.3390/ijms25189981_

Round 1
Reviewer 1 Report
Comments and Suggestions for Authors
Review of MS# ijms-3168609
The paper thoroughly examines how different preparation methods of platelet-rich plasma (PRP) impact its therapeutic properties, particularly for treating osteoarthritis of the knee (OAK). It offers valuable insights into how PRP preparation—whether used immediately after purification (utPRP) or after freeze-thawing (fPRP)—significantly influences its biological activity, especially in terms of platelet activation through the freeze-thaw process, humoral factor content, and anti-inflammatory properties.
The study effectively compares utPRP and fPRP, emphasizing how freeze-thawing alters PRP’s composition and its effects on key target cells in OAK, such as chondrocytes and synovial cells. The findings are relevant to clinical practice, as they indicate that variations in PRP preparation could lead to different therapeutic outcomes, which may improve clinical decision-making in OAK treatment.
While the paper provides a detailed analysis, there are still areas that need refinement.
Major points
1. The authors compare LPPRP and APS with and without freeze-thaw processes, but please clearly state which is recommended for OAK throughout the paper. Similarly, at the end of the introduction, clearly articulate the hypothesis and objectives of the study based on the background that led to conducting this research.
2. Please add the method for measuring the samples before freezing. Did the six healthy subjects have blood drawn and PRP prepared at the same time, and was the experiment conducted immediately? If it was not simultaneous, how were the samples stored until measurement?
3. I believe N=6 is a small sample size. Therefore, even if the P-value is significant, interpreting it may be difficult. Please add a power analysis test.
Minor points
1. In Table 1, IL-13 levels are very high in fLPRP, but there is no discussion regarding this result. How do you explain the high IL-13 levels in PRP with few white blood cells?
2. What do you think is the reason for the decrease in TGF-β in frozen LPPRP as shown in Table 1? Please add a discussion about the mechanism behind this in the discussion section.
3. I am concerned about the large SEM of GM-CSF in Table 1. Are all the values within the measurable range?
4. Please add an explanation for why gene expression in cells decreases while protein levels in the culture medium increase in Figure 2 and 3
5. The discussion is too long, so please make it more concise. Additionally, include the discussion of the points I mentioned.
I hope that my comment is very useful for the improvement of the article.
Author Response
Reviewer 1
Thank you for all of your important and insightful suggestions and questions. We have included a response to each suggestion and question below.
Major points
- The authors compare LPPRP and APS with and without freeze-thaw processes, but please clearly state which is recommended for OAK throughout the paper. Similarly, at the end of the introduction, clearly articulate the hypothesis and objectives of the study based on the background that led to conducting this research.
Response 1:
To address this concern, we recognize the importance of accumulating both basic research and clinical results before making a definitive recommendation. We hope this study contributes to that effort. We have revised the end of the introduction as follows: “In this study, we intended to investigate the anti-inflammatory effects and macro-phage polarization induced by different platelet-rich plasma (PRP) preparations.”
- Please add the method for measuring the samples before freezing. Did the six healthy subjects have blood drawn and PRP prepared at the same time, and was the experiment conducted immediately? If it was not simultaneous, how were the samples stored until measurement?
Response 2: As you correctly pointed out, blood was drawn from the six healthy subjects, and the PRP preparation was conducted simultaneously. The experiments were carried out immediately thereafter. We have revised the "Materials and Methods" section, line 361, as follows: "Half of the volume of the collected PRPs was immediately used for the experiment, while the other half of the PRP was promptly frozen at −80°C for 30 minutes (fPRPs)."
- I believe N=6 is a small sample size. Therefore, even if the P-value is significant, interpreting it may be difficult. Please add a power analysis test.
Response 3: We acknowledge the small sample size and have included the power analysis details in the Figure legends and the "Materials and Methods" section as follows:Figure legend
Figure 2 “Control group (n=5): 5 synovial cells donors, PRP group (n=30): 5 synovial cells donors, 6 PRP donors per group.”
Figure 3“Control group (n=5): 5 chondrocyte donors, PRP group (n=30): 5 chondrocyte donors, 6 PRP donors per group.”
Figure 4, 5“Control group (n=5): 5 monocyte donors, PRP group (n=30): 5 monocyte donors, 6 PRP donors per group.”
Materials and Methods section
4.4. Cell Isolation and Culture of Chondrocyte and Synovial Cells
“Control group (n=5): 5 synovial cells or chondrocyte donors, PRP group (n=30): 5 synovial cells or chondrocyte donors, 6 PRP donors per group.”
4.5. Isolation and Culture of Monocyte-Derived Macrophage and M1 Macrophages
“Control group (n=5): 5 monocyte donors, PRP group (n=30): 5 monocyte donors, 6 PRP donors per group.”
Minor points
- In Table 1, IL-13 levels are very high in fLPRP, but there is no discussion regarding this result. How do you explain the high IL-13 levels in PRP with few white blood cells?
Response 4: In LPPRP, leukocytes and erythrocytes are not completely absent (Table 1). The freeze - thawing may stimulate these cells, indicating increasing humoral factor concentrations, such as IL4, IL13, and IL1RA. This has been added to the Discussion Section.
- What do you think is the reason for the decrease in TGF-β in frozen LPPRP as shown in Table 1? Please add a discussion about the mechanism behind this in the discussion section.
Response 5: In this study, we measured only free active TGF-β, and freezing may have reduced its activity. This has been added to the Discussion Section.
- I am concerned about the large SEM of GM-CSF in Table 1. Are all the values within the measurable range?
Response 6: Thank you for your comment. All measured values were within the measurement range, but samples with high concentrations were diluted before measurement, resulting in values ranging from 112.0 to 11,211.0. This dilution contributed to an increase in the SEM.
- Please add an explanation for why gene expression in cells decreases while protein levels in the culture medium increase in Figure 2 and 3
Response 7: Thank you for your suggestion. A previous study reported that protein production is regulated by various post-transcriptional processes, and mRNA and protein expression levels do not always correlate. Additionally, since all the measured proteins are known to be present in PRP, the possibility that PRP residues remained in the wells cannot be excluded.
We added this content to the Discussion section.
- The discussion is too long, so please make it more concise. Additionally, include the discussion of the points I mentioned.
Response 8: We reorganized and shortened the Discussion section including the points you mentioned.
Reviewer 2 Report
Comments and Suggestions for Authors
Osteoarthritis of the knee significantly reduces the quality of life for many patients. From a medical/therapeutic point of view, its treatment is a very complex and difficult process. Recently, the use of platelet-rich plasma therapy has become an increasingly common therapeutic method. For this reason, the study presented by the authors on the impact of assessing the quality of the PRP preparations used depending on the method of purification and storage used is of research and practical relevance.
After careful reading of the manuscript, I find that the summary, introduction and methodology exhaustively and adequately reflect the issues raised. In addition, I believe that the selection of bibliographic items are up-to-date and correct (out of 40 bibliographic items, 14 papers have been published since 2020). Nevertheless, some adjustments are necessary to provide the reader with a better understanding of the context of the issues discussed.
However, there are few points that could improve the manuscript. My main comments relate to the discussion chapter:
1) I believe that the chapter presenting the discussion of results should be enriched with a comparison/discussion of the results obtained with known literature data in this field. The issue raised by the authors is currently discussed in the literature. In my opinion, the presented paper lacks a discussion of the presented research results in a juxtaposition with other similar data reported in the literature. In this context, I encourage the authors to complete the discussion chapter.
2) A very important aspect of any research method is the indication of its limitations. The authors have outlined the limitations of their research in the discussion section. However, I believe that the authors should present this aspect in the form of a separate chapter and discuss in more detail the impact of limitations on the validity of the data obtained in this research.
3) As before, I strongly recommend that the conclusions of the discussion section be extracted and, in addition, that the authors' discussion of research perspectives be indicated in the form of a separate chapter entitled “conclusions and future perspective”.
These suggested changes will allow the reader to better understand the manuscript presented for review.
I recommend publication after major revision.
Author Response
Reviewer 2
Thank you for all your important and insightful suggestions and questions. We have included a response to each suggestion and question below.
- I believe that the chapter presenting the discussion of results should be enriched with a comparison/discussion of the results obtained with known literature data in this field. The issue raised by the authors is currently discussed in the literature. In my opinion, the presented paper lacks a discussion of the presented research results in a juxtaposition with other similar data reported in the literature. In this context, I encourage the authors to complete the discussion chapter.
Response 1: We reorganized the discussion section including the points you mentioned.
- A very important aspect of any research method is the indication of its limitations. The authors have outlined the limitations of their research in the discussion section. However, I believe that the authors should present this aspect in the form of a separate chapter and discuss in more detail the impact of limitations on the validity of the data obtained in this research.
Response 2: Thank you for your suggestion. We have addressed the limitations of our research within the discussion section. In the journal's submission guidelines, do not require a separate section for limitations, we would prefer to retain the current structure.
- As before, I strongly recommend that the conclusions of the discussion section be extracted and, in addition, that the authors' discussion of research perspectives be indicated in the form of a separate chapter entitled “conclusions and future perspective”.
Response 3: Thank you for your suggestion. We believe that the conclusions are appropriately summarized within the discussion section. In the journal's submission guidelines, "conclusions and future perspectives" section is optional, we would prefer to retain the current structure.
Reviewer 3 Report
Comments and Suggestions for Authors
Dear Author:
Overall, the manuscript presents valuable research that is worth publishing, provided that the authors address the minor issues mentioned in the followings. The study contributes meaningful insights into how PRP preparation methods affect its therapeutic potential, which could influence future clinical practices and research.
1. The methodology is well-designed, including the use of different PRP purification kits and their effects on various cell types. The experiments are logically structured, providing a solid foundation for the conclusions drawn.
2. The manuscript acknowledges some limitations, but this section could be expanded. Specifically, discussing the small sample size and the use of peripheral blood from healthy subjects versus patients with osteoarthritis would be valuable. Additionally, addressing the potential variability introduced by different PRP kits in more detail would enhance the transparency of the study.
3. Although the discussion is thorough, it could be strengthened by a deeper exploration of the potential clinical implications of the findings. For example, how might these results influence the selection of PRP preparation methods in clinical settings? What are the next steps for translating these findings into practice?
Comments on the Quality of English Languagenone
Author Response
Reviewer3
Thank you for all your important and insightful suggestions and questions. We have included a response to each suggestion and question below.
- The methodology is well-designed, including the use of different PRP purification kits and their effects on various cell types. The experiments are logically structured, providing a solid foundation for the conclusions drawn.
Response 1; Thank you for your comment.
- The manuscript acknowledges some limitations, but this section could be expanded. Specifically, discussing the small sample size and the use of peripheral blood from healthy subjects versus patients with osteoarthritis would be valuable. Additionally, addressing the potential variability introduced by different PRP kits in more detail would enhance the transparency of the study.
Response 2; Thank you for your suggestion. We already discussed the limitation of the small sample size and the use of peripheral blood from healthy subjects in line 304 - 307.
We have revised the "Discussion" section, line 314, as follows: " Thus, while the amount of PRP injected into the joint during clinical practice differs between LPPRP and APS,
the total amount of humoral factors injected into the joint may also differ, potentially leading to different clinical outcomes. However, the same amount of PRP was added to the culture medium in this study."
- Although the discussion is thorough, it could be strengthened by a deeper exploration of the potential clinical implications of the findings. For example, how might these results influence the selection of PRP preparation methods in clinical settings? What are the next steps for translating these findings into practice?
Response 3; The next steps for translating these findings into clinical practice will be to select OAK patients based on the findings of this study, inject PRP, and evaluating the intra-articular condition and the degree of improvement in symptoms. We added this content to the Discussion section (line 322).
Round 2
Reviewer 1 Report
Comments and Suggestions for Authors
The authors sincerely responded to my comments, but their responses to the following two major points were incomplete.
In the major point 1 that I raised, I pointed out, 'please clearly state which is recommended for OAK throughout the paper,' but I could not confirm any additional clarification.
They have not sufficiently responded to the major point 3 that I raised. It seems that the author's response focuses on clarifying the sample sizes in their experimental groups, but it does not directly address my request for a power analysis. I pointed out the need for an explicit power analysis, ideally with details about the effect size, alpha level, and power used in their calculation. It helps determine if the sample size is adequate to draw reliable conclusions from the data.
Author Response
Reviewer 1
Thank you for all of your important and insightful suggestions and questions. We have included a response to each suggestion and question below.
The authors sincerely responded to my comments, but their responses to the following two major points were incomplete.
In the major point 1 that I raised, I pointed out, 'please clearly state which is recommended for OAK throughout the paper,' but I could not confirm any additional clarification.
Response:
Thank you for your comment. To clearly state which is recommended, we believe that data of clinical research is necessary, it is not possible to specify a recommendation based solely on the results of this study. Therefore, no changes have been made to the text.
They have not sufficiently responded to the major point 3 that I raised. It seems that the author's response focuses on clarifying the sample sizes in their experimental groups, but it does not directly address my request for a power analysis. I pointed out the need for an explicit power analysis, ideally with details about the effect size, alpha level, and power used in their calculation. It helps determine if the sample size is adequate to draw reliable conclusions from the data.
Response:
I apologize for misunderstanding your previous question, and I have revised the manuscript accordingly.
In this study, due to the use of different cell types, we based our effect size on previous research, setting it at 0.06. The alpha level was set at 0.05, and the target power was set at 0.80. Based on these settings, we calculated the required sample size and determined that 5 to 6 donors per group would be optimal.
We have incorporated these details into the revised manuscript in Materials and Methods section.
Reviewer 2 Report
Comments and Suggestions for Authors
Thank you to the authors for addressing my comments. I believe the revised manuscript improved in clarity. Therefore, I recommend this article for publication.
Author Response
Thank you to the authors for addressing my comments. I believe the revised manuscript improved in clarity. Therefore, I recommend this article for publication.
Response:
Thank you for your valuable comments and reviews. I would like to express my sincere gratitude.
Round 3
Reviewer 1 Report
Comments and Suggestions for Authors
Thank you for your reply. I appreciate your sincere response to my question. I wish you continued success in your research.